# Welcome to the Dark Side: Analyzing the Revenue Flows of Fraud in the Online Ad Ecosystem

## Abstract

The online advertising market has recently reached the 500 billion dollar mark. To accommodate the need to match a user with the highest bidder at a fraction of a second, it has moved towards a complex, automated and often opaque model that involves numerous agents and intermediaries. Stimulated by the lack of transparency, but also the enormous potential profits, bad actors have found ways to circumvent restrictions, and generate substantial revenue that can support websites with objectionable or even illegal content.

In this work, we evaluate transparency Web standards and show how shady actors take advantage of gaps to absorb ad revenues while putting the brand safety of advertisers in danger. We collect and study a large corpus of thousands of websites and show how ad transparency standards can be abused by bad actors to obscure ad revenue flows. We show how identifier pooling can redirect ad revenues from reputable domains to notorious domains serving objectionable content and that the phenomenon is underestimated by previous studies by a factor of 15. Finally, we publish a Web monitoring service that enhances the transparency of supply chains and business relationships between publishers and ad networks.

**ACM Reference Format:**
Anonymous Author(s). 2024. Welcome to the Dark Side: Analyzing the Revenue Flows of Fraud in the Online Ad Ecosystem. In . ACM, New York, NY, USA, 14 pages. https://doi.org/10.1145/nnnnnnn.nnnnnnn

## 1 Introduction

Like any domain of economy in which hundreds of billions of dollars are moved [19, 46], it is no surprise that with the widespread adoption of programmatic advertising, there has also been a surge in fraudulent activities. Considering the impersonal nature of the programmatic ad transactions, the complex and often opaque supply chains, the big number of intermediaries (who benefit from fraudulent traffic [27]) and the ecosystem's reliance on easily fiddled metrics [85], it is apparent that digital advertising constitutes a very vulnerable and very lucrative opportunity for fraudsters.

Studies show that one out of every three dollars spent by advertisers is wasted due to ad fraud [80], when according to Google, 56% of impressions served across its advertising platforms are not viewable for the users [82]. In fact, the global cost of ad fraud is expected to continue growing exponentially to $100 billion [76].

Recent examples of sophisticated ad fraud reveal that disinformation websites manage to receive ads (and revenue) from respectable companies that would dread even the thought of seeing their brand name next to fake news [18, 23]. Similarly, during the 2016 U.S. election campaign, hundreds of websites were created to spread fake news via click-bait headlines and generate massive ad revenue [25]. In 2018, the 3ve botnet [58] was dismantled, consisting of 1.7 million infected PCs and 10,000 fake sites. It was generating 3-12 billion daily bid requests, using over 60,000 seller IDs, thus receiving ad placements which cost a whopping $29 million.

Of course, we are not the first to identify that the lack of transparency in the supply chain creates the perfect field of action for fraudsters. There were several attempts from policymakers and stakeholders to shed light to these processes [24, 37, 44], but the mechanisms deployed were either inefficient or not widely adopted [26, 28, 39, 81]. In [6, 65, 84], authors highlight the lack of transparency in ad ecosystem, showing that inconsistencies encumber automated processes. In [86] authors measure the efficiency of brand safety tools and conduct an initial study of the prevalence of dark pooling in misinformation websites. However, since it was limited to a much smaller number of websites, it significantly underestimated the magnitude of the problem by more than an order of magnitude (in this work we show that the average dark pool size is larger than what was reported by a factor of 15).

In this work, we shed light on the techniques that bad actors deploy to abuse the online advertising ecosystem and absorb ad revenue. We highlight how current Web standards are regularly misused and how the lack of compliance with standards leaves room for loss of advertisers' money. We show that both publishers and resellers pool their identifiers together to share the ad revenue they generate from unsuspecting advertisers, thus posing a significant risk to their brand safety. We also discover that ad brokers disguise themselves as publishers claiming a larger portion of the advertisers' budget, thus making the supply chain even more opaque.

The contributions of this work are summarized as follows:

(1) We conduct the first, to our knowledge, large-scale systematic study[1] of state-of-the-art ad transparency standards across more than 7 million websites and discuss their effectiveness. We find that these standards are regularly misused since there is no verification of proper implementation, allowing bad actors to obscure ad revenue flows.

(2) Our findings show that Web publishers are able to circumvent restrictions of ad networks via *identifier pooling* and monetize misinformation or pirated content. We estimate that such bad actors are able to absorb thousands of dollars from the advertising ecosystem. Contrary to previous work [86], we show that the average dark pool size is larger than what was reported by a factor of 15, thus exposing the

---

[1]We make the source code of our tools and our datasets publicly available to support further research. See Appendix A.

real magnitude of dark pooling and show that it is more than an order of magnitude of what was thought before.

(3) We uncover 30 cases of ad resellers that disguise themselves as content owners (i.e., Web publishers) in other ad networks and abuse ad-related standards to increase their profits. We show that some of these resellers work with almost 200 objectionable or even illegal websites in total, and that their behavior does not change in a 7-month period.

(4) We build and publish AdSparency: a Web monitoring service that aims to enhance transparency in the ad ecosystem. By using data from millions of domains, it provides statistical information about the supply chains and a set of investigative tools to discover and analyze business relationships among publishers and ad networks.

## 2 Background

**ads.txt:** In the past, counterfeit inventory was a common problem with advertisers not knowing if their ads indeed appeared where they paid to [41]. Fraudsters would sell ad inventory that belonged to completely unrelated websites [6]. The ads.txt specification was introduced by the Internet Advertising Bureau (IAB) Technology Laboratory to prevent bad actors from selling ad inventory of websites without authorization [44]. An ads.txt file is a text file placed at the root of a domain. In this file, publishers explicitly disclose the entities (i.e., accounts) that are authorized to sell the ad inventory of the respective domain. An example ads.txt file is shown in Figure 1. Each record has comma-separated fields and authorizes a specific entity to sell ad inventory. The mandatory fields of each record are: (i) the domain of the ad system that bidders connect to, (ii) an identifier that uniquely identifies that account of the seller within the ad system, (iii) the type of the account. The type of the account can be either DIRECT, indicating that the web publisher directly controls the advertising account, or RESELLER, indicating that the publisher has authorized a third party to manage and (re)sell the ad inventory of the website. The ads.txt mechanism does not combat the entire spectrum of ad fraud and it relies on the assumption that *all involved entities respect the specification.* Supply-Side Platforms (SSPs) are expected to ignore inventory they have not been authorized to sell and, Demand-Side Platforms (DSPs) are expected to not buy ad inventory from unauthorized sellers.

**sellers.json:** The IAB Tech Lab introduced sellers.json files to increase the transparency of the ad ecosystem [42]. It is supplementary to ads.txt files and helps discover and verify the entities involved in ad inventory selling. Along with ads.txt, sellers.json files oppose ad fraud. Each advertising system (i.e., SSP) is expected to publish a sellers.json file, explicitly listing all registered ad inventory sellers. Each sellers.json entry (Figure 1) contains an identifier that uniquely represents the seller within the respective ad system. This is the same ID that websites disclose in their ads.txt file. Optionally, the name of the legal entity, which generates revenue under the given ID, is also specified. Each entry must specify the type of seller as one of: (i) PUBLISHER, indicating the domain's ad inventory is sold by the domain owner and that the ad system directly pays the owner, (ii) INTERMEDIARY, indicating that ad inventory is sold by an entity that does not own it but acts as an intermediary to sell it, or (iii) BOTH, when the account is both types.

```
google.com, pub-9435010515680227, DIRECT, f08c47fec0942fa0
rubiconproject.com, 20910, DIRECT, 0bfd66d529a55807
pubmatic.com,157150,RESELLER,5d62403b186f2ace
openx.com,540191398,RESELLER,6a698e2ec38604c6                    ads.txt

{
    "seller_id": "pub-9435010515680227",                       sellers.json
    "seller_type": "PUBLISHER",
    "name": "Quora Inc",
    "domain": "quora.com"
},
{
    "seller_id": "pub-6806024782369214",
    "is_confidential": 1,
    "seller_type": "PUBLISHER"
}
```

**Figure 1: Snippets of the ads.txt file served by *quora.com* and sellers.json file served by *Google*. Each record represents a business relationship.**

## 3 Methodology

We provide an overview of our methodology in Figure 2. To collect our dataset, we fetch and analyze ad-related files served by publishers and ad networks. We implement two crawlers located in an EU-based institute and visit more than 7M distinct domains during February-March 2023. First, we utilize the open-source ads.txt fetching and parsing module offered by IAB [43] to crawl and collect ads.txt files served by millions of websites. In Section 4, through an offline analysis we investigate **Identifier Pooling**: publishers that share their identifiers with unrelated websites to share revenue. We study the characteristics (e.g., size and composition) of pools of websites and demonstrate how this technique can fund objectionable content, such as fake news. In Section 5, we develop a recursive crawler of sellers.json files in order to discover **Hidden Intermediaries**: ad networks that masquerade themselves as publishers. We perform a graph-based analysis to uncover the business relationships across ad networks and find that deceitful ad systems can forward ads to unknown entities while charging advertisers higher prices. We make both of our crawlers publicly available in [4] (see Appendix A). Finally, in Section 6, we utilize the collected data and present, **AdSparency**: a Web service that illustrates the state of the online ad ecosystem, and provides stakeholders with investigative tools to uncover business relationships among Web entities. We discuss ethical aspects of our work in Appendix B.

## 4 Identifier Pooling

Assume a website that sells books (e.g., *example.com*) registers for an advertising account with Google. *example.com* will be reviewed to ensure that it does not violate any of Google's terms and eventually be granted a publisher ID so that it can display ads. Now assume an affiliated website (e.g., *fakenews.com*) that is notorious for publishing misinformation. Since *fakenews.com* has attracted negative attention for its articles, popular advertisers have pulled away from it. Thus, while *example.com* receives ads from *popular-brand.com* via programmatic ad processes (e.g., RTB, Header Bidding), *fakenews.com* gets ads from *click-bait.com*. To increase the quantity but also the quality (and price) of the ads it receives, *fakenews.com* can make a backroom deal with *example.com* to pool their ad inventory, in exchange for a small fee that *example.com* gets for sharing its publisher ID [29]. By simply putting the publisher ID of *example.com* in the ads.txt file served by *fakenews.com* and using it in bid requests, a "dark pool" is formed. The revenue generated

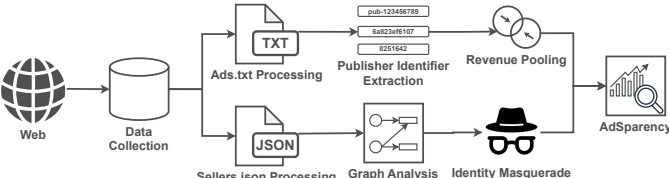

**Figure 2: Overall methodology of present study.**

from all advertisers (both *popular-brand.com* and *click-bait.com*) will wind up at the same publisher account.

Pooling ad inventory keeps both advertisers and ad networks in the dark regarding where their money flows to. Advertisers that appear on *example.com* will inadvertently have their money flow towards *fakenews.com* or other undesirable websites. To make matters worse, *fakenews.com* can use the shared ID and lie about the origin domain in the bid request, in order to directly get ads, and thus revenue, from popular advertisers. Because of pooling, almost 1 billion ad impressions were attributed to just 30 websites [28]. In fact, Breitbart, an infamous misinformation website, was using this technique to bypass block lists and generate ad revenue [22, 29].

Even though ads.txt files were introduced to tackle fraud, the lack of transparency makes it difficult to prevent revenue from being funneled to unrelated websites. Pooling identifiers is not necessarily in violation of the standard or an abuse of the ad ecosystem. However, from its early stages, the ads.txt specification was criticized and there were speculations that ad-related companies facilitate dark pooling by forcing multiple publishers to use the same identifiers [68, 69]. Unsuspecting website publishers were instructed to declare identifiers they do not control in their ads.txt files [66, 74], and had their ad inventory pooled with completely unrelated websites [22]. Advertisers rest assured that their money funds specific websites while, due to identifier pooling, it can flow towards unknown entities, threatening their brand safety. As a result, simple block lists [10, 79] are no longer sufficient to ensure that advertisers do not fund objectionable content. Advertisers would now need to block specific publisher IDs, something non realistic because it is impossible to know where each identifier is used.

Of course, shady websites can copy identifiers from other websites without permission to falsely indicate that a popular ad network is an authorized seller of its inventory. This can boost the website's reputation with other ad networks, increase their ad inventory value or even bypass review policies. There exist websites that declare identifiers in their ads.txt files that the corresponding ad networks do not even acknowledge (Appendix E). There is already evidence that questionable websites can monetize their ad inventory using this technique [86]. Finally, if a less popular ad network observes that a website uses publisher IDs from popular ad networks (e.g., Google), they might not perform a manual review.

### 4.1 Data Collection

To study Identifier Pooling, one must have access to the identifiers publishers declare in their ads.txt files. To that extent, we utilize IAB's official ads.txt crawler [43]. We keep the implementation as-is and only change the user-agent header so that we are not blocked by websites. We extract almost 7 million websites from the Tranco

list [45] that aggregates the ranks of domains[2] and crawl them during February 2023. The crawler successfully fetched the ads.txt files of 456,971 domains and extracted 81,985,768 valid entries that follow the specification [44]. In these entries we detect 591,546 distinct DIRECT publisher identifiers. DIRECT identifiers indicate that the content owner directly controls the account responsible for selling a website's ad inventory. We focus on such identifiers since they indicate a direct business relationship between the ad network and the publisher [6, 44]. Sharing them with unrelated entities is in violation of the specification. RESELLER accounts are expected to handle the ad inventory of multiple websites, form heterogeneous pools and redistribute ad revenue [44]. Consequently, such identifiers are excluded from the analysis of this work.

### 4.2 Pools Composition

We analyze the collected ads.txt records and, in Figure 3, we observe that popular ad networks are equally represented in websites and share a similar portion of the market. Google is the only exception, since it evidently dominates the market. 92% of websites that serve an ads.txt file contain at least one DIRECT publisher ID issued by Google for the monetization of the website's content.

We define that a DIRECT identifier is used to form a pool when the same identifier is used in more than one website. Contrary to previous work [86], we follow a more strict definition of pools by focusing only on DIRECT identifiers. Sharing a DIRECT identifier does not inherently indicate an abuse of the ad ecosystem. Websites operated by the same entity are allowed (even expected) to use the same identifier across websites [62]. Pooling violates the specification when an ID is shared among unrelated websites. Identifiers might also be shared across websites due to intermediary publishing partners [32, 62]. These third-party services manage ad inventory of multiple publishers to optimize their ad revenue. However, they should not register their IDs as DIRECT and then distribute them to theirs clients since they do not own the ad inventory.

Overall, we find 185,535 distinct pools. In Figure 4, we plot the most popular ad networks whose DIRECT identifiers are used to form pools. We plot (i) the percentage of all detected pools formed by identifiers of a specific ad network (green line), and (ii) the percentage of identifiers that each ad network has issued and are used to form pools (black bars). First, we observe that all ad networks that dominate the market (Figure 3) also allow their identifiers to form pools. Most prominently, direct identifiers issued by Google form 35% of all pools in our dataset, while Taboola's identifiers form 6% of all pools. This is not innately damaging for the ecosystem. Nonetheless, we also observe that over 70% of the DIRECT identifiers from 4 popular ad networks are used to form pools. This suggests that not all ad networks properly use ads.txt relationships or that they do not properly monitor how they identifiers are used.

Inspired by previous work regarding monetization of fake news websites (e.g., [5, 8, 63]), we explore if objectionable websites participate in identifier pooling. We form two lists of objectionable websites. First, we make use of MediaBias/FactCheck (MBFC) [11], an independent organization that detects bias of information sources and extract 1,163 misinformation websites that are extremely biased and often promote propaganda or have failed fact-checks. We also

---

[2]https://tranco-list.eu/list/998W2/full

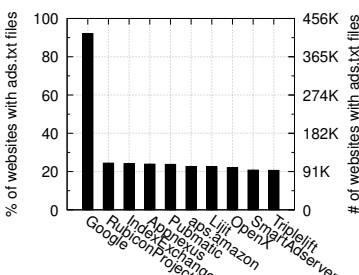

Figure 3: Popular networks whose identifiers are used by websites to monetize their content based on `ads.txt` records.

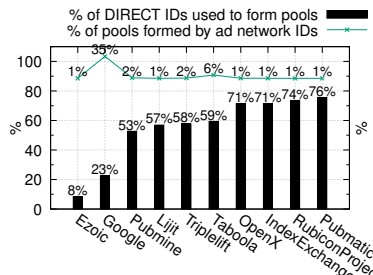

Figure 4: Popular networks whose identifiers are used to form pools of websites sharing the same publisher identifier.

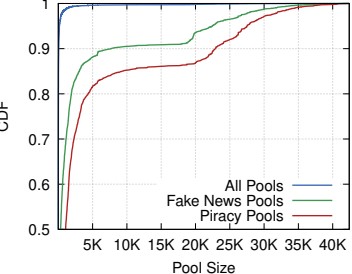

Figure 5: Distribution of pool sizes (i.e., number of websites) based on the types of websites they contain.

form a list of known Web piracy websites. We utilize NextDNS' Piracy Blocklist [57] and focus on 1,395 websites in the "torrent" and "warez" categories. We make both lists publicly available [4]. We acknowledge that other types of objectionable websites might also deploy identifier pooling, but focus on misinformation and piracy websites because these websites are prone to committing ad fraud (e.g., [14, 55, 86, 87]). We find that there are 211 fake news websites and 121 piracy websites that participate in pools of various ad systems. Interestingly, there are over 5,000 and over 2,000 pools that contain at least one misinformation or piracy website, respectively. This suggests that there are numerous benign websites found in the same pool (sharing revenue) with objectionable websites and that, this can happen without their operators knowing.

We study the size of pools based on the type of websites they contain. In Figure 5 we illustrate the distribution of pool sizes for (i) all pools, (ii) pools that contain at least one fake news website, and (iii) pools that contain at least one piracy website. We observe that general pools are smaller than pools that contain objectionable websites. In fact, both the mean and the median general pool size is much smaller than for pools containing fake news or piracy websites. For general pools, the median pool size is 4, while the mean is ~122. Publishers often operate a few websites and use the same identifier in order to monetize them and keep track of their traffic [62]. On the other hand, the median and mean pool sizes for pools with fake news websites are 336.5 and ~2,975, while for piracy pools they are 645 and ~4,915 respectively.

We perform two-sample Kolmogorov–Smirnov tests between the distribution of all pools and fake news or piracy pools, respectively, to verify this observation. We find that the KS test is 0.6 with a p-value 5.0815e-320 for the comparison between all pools and fake news pools, and 0.77 with p-value 1.86e-320 between all pools and piracy pools. The two-sample KS tests indicate that fake news and piracy pools are significantly different from the identifier pools formed on the Web in general. This suggests that identifiers might be falsely registered as `DIRECT` and that there is no single entity that directly controls thousands of websites, but rather that objectionable websites tend to cluster inside big pools and use the same identifier as multiple other websites to be served ads [54].

### 4.3 Ecosystem Abuse

We set out to explore if websites do in fact abuse the advertising ecosystem by sharing identifiers with unrelated websites. To verify this hypothesis, we attempt to establish the organization that

| Issuing Ad Network | Identifier | Issued To | Number of Websites |
|---|---|---|---|
| *conversantmedia.com* | *100141* | 33Across | 42,412 |
| *vi.ai* | *987349031605160* | OutBrain | 41,044 |
| *adform.com* | *1942* | Rich Audience Technologies SL | 40,702 |
| *onetag.com* | *5d4e109247a89f6* | ConnectAd Demand | 40,660 |
| *lijit.com* | *244287* | ConnectAd Realtime | 40,587 |
| *indexexchange.com* | *190906* | ConnectAd Realtime | 40,061 |

Table 1: `DIRECT` identifiers used in numerous websites.

operates each website. Identifier pooling is not inherently harmful but problems arise when websites share the same `DIRECT` IDs with other unrelated websites. To that extent, we utilize the WHOIS [20] protocol to find the owner organization of a domain, as performed in previous work [9, 71]. We query registrars for each domain inside a pool and extract the domain owner (i.e., registrant). To tackle the WHOIS privacy service offered by registrars [49], we manually review all extracted records and create a list of 60 keywords that signify records have been redacted for privacy concerns. We make this list public [4] and exclude respective records from further analysis. We retrieve the owner organization of 3,981 websites.

We process each pool in our dataset and for each website in it we extract the owner organization from the respective WHOIS record. We perform a case-insensitive matching of organization names in order to overlook any typos during WHOIS registration. Additionally, we exclude organization names less than 3 characters long because we believe they do not represent actual entities. To our surprise, we find almost 15,000 distinct dark pools. These are pools of websites owned or operated by different organizations but sharing at least one `DIRECT` identifier, thus violating the `ads.txt` standard. Such pools distort the ecosystem making it practically impossible to understand where advertiser money is directed. To make matters worse, 59.6% of pools with at least one misinformation website are dark pools. For pools with piracy websites, this percentage rises to an astounding 82%. Even though we were able to extract the owner of a small portion of websites due to privacy restrictions, our findings are a lower limit of what happens in the wild. Private WHOIS records do not affect the correctness of our findings, making them representative of the Web.

Next, we investigate how the behavior of pooling changes based on the size of the ecosystem one studies. We incrementally analyze larger sub-portions of the ecosystem based on the popularity of the websites. For each sub-portion, we only study websites ranked up to that specific popularity, based on the Tranco list [45]. We use

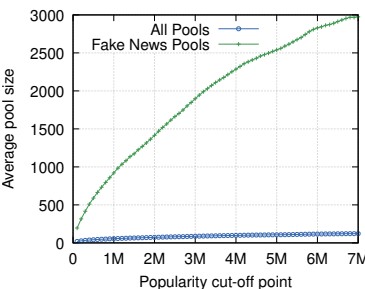

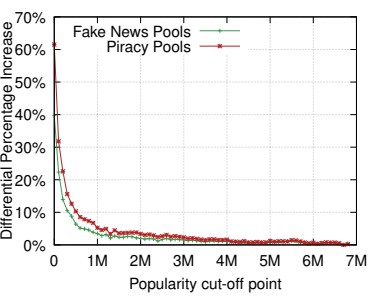

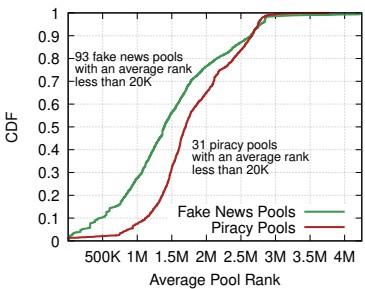

**Figure 6: Average pool size for different popularity cut-off points. A larger set of websites reveals more aggressive pooling.**

**Figure 7: Differential increase of average pool size for different popularity cut-off points.**

**Figure 8: Average rank of pools containing at least one objectionable website. A small rank value signifies a popular site.**

intervals of 100K ranks and plot in Figure 6 the average pool size for general pools, and pools with at least one fake news website. We observe, that even though the average pool size for general pools has minimal increase, the average pool size of pools with fake news websites increases substantially. We compare the average fake news pool size of our entire dataset (i.e., rightmost data point in the green line of Figure 6) with the average fake news pool size of the top 100K most popular websites (i.e., leftmost data point in the green line) and find that it is 15 times larger.

It is evident that by studying a greater portion of the Web, we are able to better understand the phenomenon of identifier pooling and that focusing on only popular websites, as done in previous work [86], underestimates the problem by more than an order of magnitude. Additionally, in Figure 7, we plot the differential percentage increase of the average pool size from one popularity cut-off point to the next. We find that after the top 1M most popular websites, this increase reaches a plateau, indicating that after this point, pools remain somewhat rigid. This finding increases our confidence that we are able to study the problem of identifier pooling in its entirety. In fact, the rightmost points of this figure represent an increase of less than 1%. This behavior is expected, as one could argue that it is not beneficial to include less popular websites to pools, since they will not boost the aggregated revenue.

Finally, our findings suggest that there is no correlation between the popularity of a website and the extent of the mis-registered identifiers. The Pearson correlation coefficient between the Tranco rank of a website and the number of pools it participates in is -0.05 (p-value 9e-129), suggesting that there is no meaningful correlation between website popularity and abuse intensity. However, there exist 40 highly popular websites (ranked in the top 50K most popular websites worldwide) that participate in an extreme amount of distinct pools. For example, *narod.ru* is ranked 2,019th worldwide and serves an `ads.txt` file of over 3,200 `DIRECT` Google identifiers.

### 4.4 Revenue Generation

The total traffic of a website greatly affects its ad revenue. We plot in Figure 8 the average rank of pools that contain at least one fake news or piracy website (i.e., average rank of websites it contains). We find a lot of pools with a very high average popularity rank, meaning that the websites in these pools attract heaps of visitors. Surprisingly, we find that there are 93 fake news pools and 31 piracy pools with an average rank less than 20K, suggesting that they

contain extremely popular websites, have loads of daily visitors and are able to generate big amounts of ad revenue. Even if it is split across multiple websites or reduced by handling fees, it can still be a significant income for publishers of illicit or unethical content.

To verify this, we extract from SimilarWeb [48] the sum of all visits during September 2023 for websites in highly popular pools (i.e., average pool rank less than 20K). Indeed, we find that the median website in these pools had 6.7M visitors during September 2023 and that there are 6 websites, which totaled over 100M visitors. In an attempt to translate this vast amount of visitors to ad revenue, we also extract the estimated annual revenue for each website. We were able to extract revenue data for 16 websites and their estimated annual revenue is in the millions. In fact, the lowest estimated revenue is 2M-5M\$, with two websites found in fake news pools having an estimated annual revenue of over 1B USD.

It is evident, that if websites have the right to get a share of the revenue from each of the publisher IDs they use, they can generate revenue from multiple sources. To better illustrate this, we plot in Figure 9 the revenue flow that originates from popular ad networks towards fake news websites, found in the most pools. When a website is part of a pool formed by an identifier of ad network $Y$, then there is a flow from that network towards the website. When a website is found in multiple pools of a specific ad network, the flow between these entities carries a greater weight. This figure illustrates only potential revenue flows. Due to the complexity of the ad ecosystem and the various entities involved, ads might be served to a website through a different route (i.e., ad network). First, we discover that there is an ad revenue flow from all popular networks towards extreme misinformation websites, indicating that misinformation websites are able to not only monetize their content, but to also generate revenue from multiple sources. We observe that the revenue flows that originate from Google and Lijit are the most prominent. In fact, fake news websites tend to participate in multiple pools formed by Lijit's identifiers.

However, it should be noted that simply copying an ID from another website does not provide any monetary benefit to the bad actor (i.e., they don't make any money out of it). In order for the bad actor to gain revenue, there has to be an agreement with the owner of the ID (e.g., associate the website with the ID in the ad-management platform). As described, this happens either through deals with other publishers or through the facilitation by ad networks. We provide examples to demonstrate the effect of identifier pooling on the ad ecosystem in Appendix D.

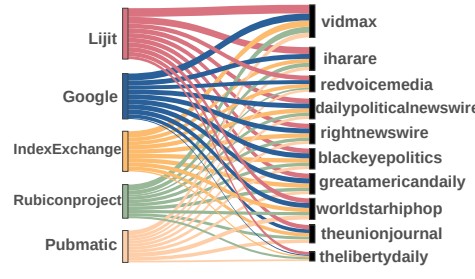

**Figure 9: Flow of revenue from the top 5 ad networks towards fake news websites that participate in inventory pooling.**

## 5 Hidden Intermediaries

It is important for advertisers to know where their ads will be rendered, not only to determine whether they reach their target audience, but to also ensure that their ads do not appear next to "bad" content. Due to the complexity of programmatic advertising, even reputable companies often have no control over where their ads appear [1]. This can be a very big hit for their reputation [78] and their brand's safety. As a result, they often prefer supplying ads directly to publishers and not via intermediaries [54]. Yet, previous studies have shown that only half of the advertising budget reaches the publishers themselves and that the other half is absorbed by intermediary entities [59]. To mitigate this, in the RTB ecosystem, advertisers tend to prioritize buying ad inventory from IDs that have been registered as PUBLISHER. This way, no intermediary entities are involved in reselling the ad inventory beyond advertisers' control and they can better track the flow of their ad spending.

Is it possible for some media companies to falsely register as a publisher in another ad network in order to abuse the ecosystem? Such media companies would masquerade themselves as publishers while, in fact, they are intermediaries that manage the ad inventory of numerous websites or publishers. As a result, *hidden intermediaries* could charge higher (i.e., Cost Per Thousand Impressions) because they pretend to be publishers while rendering ads beyond the control of the advertisers on questionable websites. Such behavior deteriorates the ecosystem's transparency since hidden intermediaries deceive buyers ther prefer shorter supply chains, and resell ad slots to their own clients. In fact, due to complexity, one third of the supply chain costs are un-attributable [59].

### 5.1 Data Collection

To study the phenomenon of hidden intermediaries, we need to keep track of the relationships between ad systems and their clients. We collect and analyse the `sellers.json` files that ad systems serve. We build a tool that visits a domain and attempts to fetch a `sellers.json` file, if it is served on the domain's root and perform a recursive crawl. That is, if we detect a `sellers.json` file, then we recursively visit all the domains listed in it and attempt to download their own `sellers.json` file. Using the Tranco list[3] as a seed to our tool, we crawl the Web on March 24, 2023. In total, we visit 7,341,165 distinct domains and detect a `sellers.json` file in 2,682 domains. A small number of domains that serve `sellers.json`

---

[3]https://tranco-list.eu/list/998W2/full

files is expected since only ad systems should publish those. From the detected files, we extract over 34 million `sellers.json` entries.

We study the state of the `sellers.json` standard to get a better understanding on how Web entities have adopted it. Unfortunately, we find that the `sellers.json` specification is not properly implemented in the wild. There are 26K publisher IDs that are declared with the wrong relationship type in `ads.txt` files and that happens for identifiers of all ad networks. Additionally, it is evident that ad networks pay little attention to the `sellers.json` they serve because users can claim any website they want (even popular ones), while there exist domains that serve a `sellers.json` file of a completely unrelated ad network. Finally, we discover ad networks that dilute the transparency of supply chains by hiding the entities that own publisher IDs. We analyse such violations in Appendix E.

### 5.2 Ecosystem Abuse

We classify an ad network $X$ as a hidden intermediary if (i) it serves a `sellers.json` file, and (ii) has at least one named client (i.e., at least one non-confidential entry in their `sellers.json` file), and (iii) $X$ is registered in another ad network $Y$ as a PUBLISHER, and (iv) $X$ is registered in another ad network $Z$ as an INTERMEDIARY. This inconsistent behavior might be credited to human error, or it might suggest mischievous motives. There have been cases where middlemen were mislabeled as publishers [53] (i.e., hidden intermediaries), and were working with popular disinformation websites without the advertisers or the issuing ad network having any control [56].

We attempt to discover hidden intermediaries through the inferred relationships from the collected `sellers.json` files. To increase the confidence of our findings, we only retain "verified" ad networks. IAB Tech Lab's `ads.txt` crawler [43] contains a list of popular ad system domains that they take into consideration when processing `ads.txt` records. This step is necessary to increase confidence that entities are indeed ad brokers due to the discrepancies and deviation from the specification (Appendix E).

We discover 33 ad networks that have been falsely registered as publishers even though they are in fact middlemen, and simultaneously, represent hundreds or even thousands of actual publishers. For example, we find that Smaato, a popular ad platform that manages the ad inventory of over 1 thousand publishers, is a hidden intermediary. In the `sellers.json` files that *keenkale.com, lkqd.com* and *adingo.jp* serve, Smaato is wrongfully presented as a publisher. This is a major issue for brand safety since Smaato is able to receive bids through these ad networks as a publisher and either charge for higher CPM, or obfuscate the ad chain. We illustrate in Figure 10 the top cases of verified hidden intermediaries. Kiosked displays the most extraordinary behavior, hiding as a publisher in 67 other ad networks. Finally, we uncover that "The Publisher Desk", "Freestar", "Aditude" and "Next millennium Network" are all still hidden intermediaries even though they have attracted attention for exactly this practice in the past [56].

We acknowledge that this behavior can arise from simple human errors when forming the `sellers.json` file or when registering for an ad account. To address this, we investigate how the identifiers issued to hidden intermediaries are used. We find 1,860 cases where an ad network has registered as a publisher in a different ad network, was issued an identifier and then distribute this DIRECT identifier

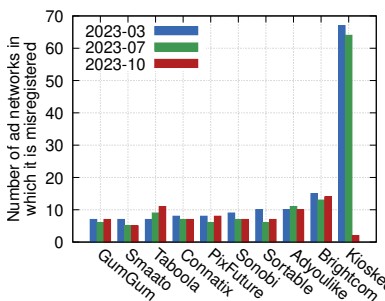

**Figure 10: Extreme cases of ad networks that masquerade as publishers in other ad systems.**

to more than 10 websites. For instance, Kiosked, was registered as a publisher to *yahoo.com* and was issued the identifier 56848. However, there are almost 1,500 websites (even popular ones) that disclose this identifier as DIRECT in their ads.txt.

To decrease the possibility of a human-error when forming sellers.json files, we perform a temporal analysis and re-crawl the same list of 2,600 domains with sellers.json files from Section 5.1 on July and October 2023 (Figure 10). We find that there are 34 "verified" ad networks hiding as publishers in July 2023, rising to 37 in October 2023 (increased by 4 during the past 7 months). Intermediaries who are registered as publishers the most, have not greatly changed their behavior over the period of 7 months. In fact, there is a significant change only in the case of Kiosked. We recognize that not all cases of hidden intermediaries suggest a mischievous motive or active attempts to commit fraud. However, it is evident that ad standards do not work. We highlight that not only is the ecosystem unable to provide transparency and confidence, it also enables bad actors to abuse it [56].

### 5.3 Indirect Clients

A very important issue with hidden intermediaries is that they manage the ad inventory of numerous publishers and they are able to connect these publishers to the advertisers that mistakenly bid to the hidden intermediary's identifier. The problem is that these publishers might have not been vetted, and advertisers might inadvertently fund publishers that do not follow regulations or have bad reputation. We study the clients of hidden intermediaries; entities that the ad network discloses inside the sellers.json as certified ad inventory sellers. Using the lists of fake news and piracy websites (see Section 4), we find that hidden intermediaries manage the ad inventory of multiple questionable websites that advertisers most likely would not want to advertise in [7].

Interestingly, we discover that RevContent, a popular ad system, is hiding as a publisher in other ad networks (e.g., *mowplayer.com*), while at the same time managing the ad inventory of 33 distinct fake news websites. RevContent is popular among fake news websites [63] and fake news websites rely on Revcontent to generate revenue [36]. In total, we find 8 ad networks that manage at least 10 fake news websites each. Similarly, we discover that *reklamstore.com* and *automattic.com* managing the ad inventory of 3 websites of our piracy list each. To make matters worse, we discover 4 hidden intermediaries, *teads.tv*, *nsightvideo.com*, *monetizemore.com* and *ad-colony.com* that have approved the monetization of 4 illegal websites

through their ad platforms. These websites have been marked as illegal gambling sites by the Belgian Gaming Commission [16] and advertisers wouldn't want to see their brand next to such content.

In total, we find that there are 167 fake news websites, 19 piracy websites and 4 illegal websites that are managed by hidden intermediaries. All these websites are clients of hidden intermediaries that can charge higher CPM rates and forward ads to websites operated by unknown entities. This can have negative effects for both the advertisers and the ad ecosystem. Advertisers have their reputation at risk if their brand name appears next to misinformation or illicit content, while on the other hand, the entire ad ecosystem is funding objectionable content without anyone knowing.

To translate how these findings can damage the ad ecosystem, we study these websites in terms of internet traffic. SimilarWeb [48] provides website traffic data for 181 websites whose ad inventory can be catered by hidden intermediaries. Additionally, we extract their popularity rank from the Tranco list[4]. We find that the median website is ranked 152K, while on average, these websites are ranked just bellow 400K and have 10M distinct visitors per month. Each visitor accounts for 2.33 pages per visit and an average visit of 2:40 minutes, resulting in multiple ad renders. We estimate that clients of hidden intermediaries generate an average yearly revenue of 36K$ and that hidden intermediaries can cost advertisers 5.3M$ annually. We provide a detailed description of this analysis in Appendix F.

### 6 AdSparency Service

To enhance the transparency in the online advertising ecosystem, we develop and publish a Web monitoring service that utilizes information extracted from ads.txt and sellers.json files, and enables (i) stakeholders (e.g.,, advertisers, DSPs, Web publishers) to better understand where their money is funneled and what content they support, and (ii) policymakers (e.g.,, IAB, WFA) to better understand ad revenue flows and business relationships. Specifically, this service periodically crawls millions of domains, aggregates files served by different domains and analyzes the corresponding entries. Then, it (i) provides important statistical information about Identifier Pooling and Hidden Intermediaries, and (ii) provides a collection of investigative tools. Specifically, AdSparency provides tools to (i) lookup identifier pooling, (ii) detect hidden intermediaries, (iii) study website partnerships, and (iv) reveal business relationships among publishers and ad networks. By crawling large sets of websites, this service can zoom out and reveal to marketers, publishers and ad agencies the bigger picture: how publisher IDs are used and what relationships are formed between the various Web entities in a global scale. We thoroughly describe the functionality and utility of AdSparency in Appendix G.

### 7 Countermeasures

**Identifier pooling:** Publisher IDs are a mechanism used in various steps of the ad-serving pipeline. We propose techniques that can help tackle dark pooling from different vantage points. First, we propose that ad networks properly review and vet their clients and ensure that the identifiers are used properly. Ad networks should have specific policies regarding ad inventory pooling and not allow third-party entities to issue or handle identifiers on their

---

[4]https://tranco-list.eu/list/5Y9NN/full

behalf [86]. Additionally, ad networks should focus more on detecting ad fraud [29] and flag or even de-monetiaze domains that have been found to participate in dark pools. Such a behavior ensures that objectionable websites will not be able to circumvent policies and get funded through advertisements. Moreover, we propose a modification of ad-related Web standards. Specifically, we propose a strict adherence to the DIRECT definition, where each DIRECT identifier will now be strictly associated with a finite set of websites, operated by the same entity. This set of websites should be explicitly and publicly stated in the sellers.json domain field. This way, all ad entities can have a clear understanding of what websites are being funded by a specific direct identifier. We also argue that there should be an upper limit on the number of websites that can be associated with a DIRECT identifier. An unlimited number of websites would defeat the purpose and result in similar problems as with the current state of the ad ecosystem. Finally, we believe AdSparency can help stakeholders gain insights on how websites are interconnected and where advertiser money could end up and enable to work towards a more transparent ecosystem.

**Hidden Intermediaries:** Intermediaries can abuse the ecosystem if they are able to disguise themselves as publishers. The most effective countermeasure is for other ad networks to strictly review their clients and ensure that their identifiers are used in compliance with the standard. Independent evaluation of the domains found in sellers.json files can also lead to the detection of hidden intermediaries [56]. AdSparency can help towards this direction. Finally, the main issue of hidden intermediaries is that there is no clear understanding of where advertiser money is directed to, who benefits from this revenue and to what extent. We urge towards a more transparent ecosystem and propose that ad networks strictly adhere to the ads.txt and sellers.json standards and that they avoid using the confidentiality flag in sellers.json files unless strictly necessary. At the very least, we argue that the domain that has registered for an identifier should be a mandatory field and always be visible. We support IAB Tech Lab's work towards new versions of ad standards that explicitly disclose who owns and who manages ad inventory [40]. Such modifications can reveal the true relationships between ad entities and stop the need for hidden intermediaries. Additionally, we argue that ad exchanges should only accept ad networks with a valid sellers.json file.

## 8  Related Work

The advertising ecosystem has been thoroughly investigated including the cost of rendered ads [64], the advertising value of users [60] and how advertisers are paired with publishers [50]. In [6], authors performed a study of ads.txt files standard and its adoption during a 15-month period and found violations of the standard, and that they are not fully integrated in the ad ecosystem. In [31], the authors studied the advertising ecosystem and Google services and focused on how revenue is generated across aggregators. In [86], authors studied the prevalence of dark pooling. They utilized ads.txt and sellers.json files to show how misinformation websites are able to deceptively monetize their content and how dark pooling circumvents brand safety. Similar to our work, they find that misinformation websites use dark pooling to abuse the ecosystem and monetize their content. Contrary to this paper, their study is limited

to a smaller number of websites and significantly underestimates the prevalence of the problem by more than an order of magnitude.

The research community has dedicated significant effort to discover, study and mitigate ad fraud [3]. Bad actors have devised various ad fraud techniques including "Click-Jacking" [2, 21, 30, 88] or content injection [75, 83]. In [52], authors demonstrated an ad fraud attack were malicious publishers pollute the profile of visitors, compelling advertisers to pay more to reach users. Popular online platforms have been used to either serve political ads that bypass policies [67] or even to generate ad revenue from copyrighted content [14]. Similarly, in [51], authors demonstrated that not all online video platforms are able to discover ad fraud. In [77], authors established how automated farms of real smartphone devices can be used to commit ad fraud and generate substantial revenue, while in [89], authors studied ad fraud on Android applications focusing on fake click actions. In [47], authors explored ad fraud attacks that can take place in WebVR applications.

In [5], authors discussed how the lack of understanding and control advertisers have regarding where they ads appear, enables fake news websites to generate revenue. In [63], authors utilized ads.txt and sellers.json files to reveal the business relationships between fake news websites and ad networks, showing that popular ad networks inadvertently facilitate the proliferation of fake news content. In [87], authors studied problematic ads and their prevalence across news and misinformation websites, as well as the ad platforms that serve them. In [8], authors explored the advertising market and found that even though fake news publishers interact with fewer ad servers, they still rely on credible ones to monetize their traffic. Similar results were found in [36], where the authors studied how Web infrastructure supports misinformation and hate speech websites. In [61], the authors studied a novel technique that bad actors deploy to mislead advertisers into paying for ads next to pirated or illicit content. Finally, various works have studied how the quality of content can affect the brand reputation of advertisers [7, 73].

## 9  Conclusion

Due to the complex and often opaque supply chains, and the big number of intermediaries who benefit from inflated ad traffic, it is apparent that digital advertising constitutes a very vulnerable and lucrative opportunity for bad actors. In this work, we present and study the mechanisms that bad actors deploy in order to bypass restrictions policies of ad networks. We show how publishers of websites with questionable or even illegal content are able to increase their ad revenue by pooling their ad identifiers together with the ones of reputable websites. We also study the sellers.json standard and show that not only it is not properly used on the Web, but also that some intermediary ad brokers abuse it in order to masquerade as publishers and make money from ads that they then push towards objectionable websites. We establish that the ads.txt and sellers.json standards are not enough to prevent ad fraud and are constantly misused or abused. We believe that the findings of this work can help make the ad ecosystem more transparent, motivate regulators and provide stakeholders with the tools they need to stop the proliferation of objectionable content through ad fraud.

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

## A Data & Code Availability

To support and enable further research and the extensibility of our work, we make publicly available [4]:

(1) Extensive lists of misinformation websites and websites associated with pirated content.
(2) List of keywords indicating private WHOIS records.
(3) Source code of crawling tools for ads.txt and sellers.json files.

## B Ethical Considerations

This work has followed the principles and guidelines of how to perform ethical information research [38, 70]. In accordance to the GDPR and ePrivacy regulations, we do not engage in collection of data from real users, neither do we share with other entities any data collected by our crawler. We only collect and analyze information served intentionally by Web entities and is designed to be collected in a programmatic fashion according to the specifications [42, 44]. Concerning the analysis of Sections 4 and 5, we minimize our intervention on the ecosystem by designing our crawlers to be as unintrusive as possible. We contact each domain and only issue a single HTTP(S) request to fetch either the ads.txt or the sellers.json file. Additionally, the collection of ads.txt and sellers.json files was done in separate periods of time, ensuring that we only reach each domain once per month. Finally,

even though we study the advertising ecosystem, we do not interact with ads displayed in websites in any way, thus not depleting advertiser budgets.

## C  Ads.txt Specification Violation

Further studying `ads.txt` records reveals that not all entities respect the `ads.txt` specification. We discover that there are multiple ad networks that consistently re-use the same DIRECT identifier across thousands of websites. Each *adtarget.com.br* identifier is used in 4,249 websites on average. Similarly, each DIRECT identifier issued by *reforge.in* or *adriver.ru* is used in over 2,000 websites on average. These findings support our hypothesis that the `ads.txt` standard is not properly implemented. DIRECT identifiers are shared across unrelated websites, thus ruining the ecosystem's transparency. Most importantly, the DIRECT identifier 100141 issued by *conversantmedia.com* is found in 42,412 distinct websites. A DIRECT identifier should indicate that the content owner (i.e., publisher) directly controls the advertising account [44] but it seems extremely improbable that one single publisher manages the content of over 42 thousand websites. ConversantMedia explicitly states [17] that this identifier belongs to the ad network 33Across and that it is in fact an intermediary. However, even popular websites such as WikiHow and IGN list it as direct.

It is important to notice that the blame for such behavior does not always fall to the publishers themselves. It seems implausible that over 42,000 website administrators conferred with each other and reached an agreement about how to use the identifier. Similar behavior is observed for multiple other publisher IDs, as shown in Table 1. Closer inspection of these identifiers reveals that even though they are labeled and used as DIRECT, they have been issued to media companies (i.e., resellers). For example, *vi.ai* clearly states that the identifier 987349031605160 was issued to an intermediary but we find that publishers claim it as their own (i.e., DIRECT). Altogether, there are strong indications that there are multiple ad resellers that provide their own direct identifiers to their clients, having them mark them as DIRECT. It is evident that ad networks play an important role in identifier pooling. Not only some of them facilitate pooling (Figures 3 and 4), but some resellers also deliberately mislabel the ad inventory of their client publishers and abuse the ecosystem in an effort to increase their profits [22]. Rich Audience Technologies, which controls one of the largest dark pools (Table 1), has been promoting such bad practices for years [15, 54].

> **Finding:** Popular ad networks facilitate ID pooling and host resellers that purposefully mislabel their identifiers as DIRECT to form pools of thousands of websites and increase profits.

## D  Pooling of Conflicting Ad Inventory

We discover a pool of 14 websites that all disclose the same publisher ID pub-3176064900167527, issued by Google, in their `ads.txt` files. Out of these 14 websites, 3 (*sputniknews.com*, *ria.ru* and *snanews.de*) have been labeled as questionable by MediaBias/-FactCheck because they have poor sourcing and multiple failed fact checks, thus spreading misinformation. These misinformation websites would most definitely not get approved by Google, but they can use the issued identifier to by pass blocklists and to receive ads even from very respectable websites [28]. Additionally, we discover

that *inosmi.ru*, another news website, is part of the same pool, uses the same identifier, and according to SimilarWeb, achieves over 14 million monthly visits. Google's revenue calculator [35] estimates that such a website can have an annual revenue of several hundred thousand dollars. It is evident that this revenue, even if split across 14 publishers (worst-case example), is a substantial income for these publishers. The revenue that *inosmi.ru* generates indirectly facilitates the proliferation of fake news content and advertisers that appear on a legitimate news website, inadvertently support misinformation.

We also discover that the websites *newscientist.com*, *gbnews.com* and *gbnews.uk* all disclose 2 distinct identifiers issued by SpotX.tv and Sovrn as DIRECT. This suggests that these websites have two shared ad revenue wallets (i.e., accounts that collect ad revenue). Additionally, these websites seem to be unrelated and operated by different entities. They disclose a different registered office address and a different company number in their websites' terms. Consequently, they form "dark pools". GB News UK has been marked as a "conspiracy theory", "pseudoscience" and "propaganda" news source by MBFC since it has almost 10 failed fact checks [12] (almost all of them are related to COVID19). On the other hand, we discover that New Scientist is a pro-science website with very high factual reporting and high credibility [13]. By sharing a direct publisher ID, any ads that are rendered on these websites through this identifier will result in revenue being aggregated to the same account. To make matters worse, GB News can use the shared ID to directly receive ads from all sorts of brands, even popular ones. Even if this is done without the website administrators being aware (e.g., through the facilitation of ad networks [34, 86]), advertisers unintentionally have their money support a misinformation website.

## E  Sellers.json Specification Violation

### E.1  Misrepresentation

The `sellers.json` standard is complementary to the `ads.txt` standard, and together, they increase the transparency of the ecosystem and enable entities to discover the identity of ad inventory sellers. In order for this to work, publishers' `ads.txt` files need to list the ad systems they have authorized to serve ads on their websites, and ad systems need to publish a `sellers.json` file that confirms that they have reviewed the website. As already discussed, various intermediaries might want to hide the actual type of their contract with an ad network. We study the types of seller accounts declared in `ads.txt` and `sellers.json` files we have collected. When a publisher identifier is listed DIRECT in an `ads.txt` file, the respective `sellers.json` file of the ad network should list the same identifier as PUBLISHER [44]. Similarly, a RESELLER identifier in `ads.txt` should be registered as INTERMEDIARY in `sellers.json`.

We follow a graph-based approach to better understand and visualize the relationships between various domains. Specifically, every ad network is represented as a node, and if there is a `sellers.json` entry indicating an authorized relationship, we introduce an edge connecting the corresponding nodes. We analyse the collected `ads.txt` and `sellers.json` files and find that there are over 26K identifiers issued by 421 distinct advertising networks that have at least one relationship type mismatch. We detect cases of type

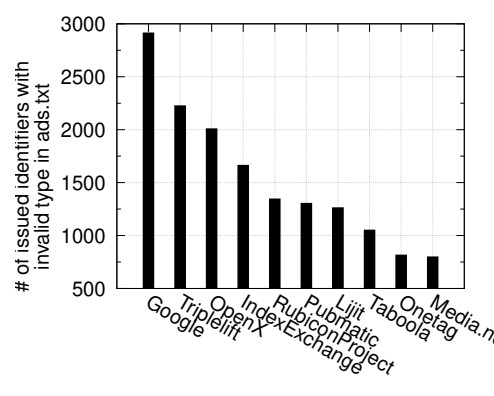

**Figure 11: Number of identifiers that have been mis-typed in `ads.txt` files as a function of issuing ad systems.**

mismatch even for popular ad networks, including Google, App-Nexus, OpenX and IndexExchange. In Figure 11, we plot the number of identifiers with a type mismatch for popular ad networks. We discover that the problem of type mismatches is not unique to a network and that all of them suffer from it. Please note that for a type mismatch, both the ad network and the publisher might be at fault. Such mismatches might derive from either a human error or from a deliberate malicious action. To justify this opinion, we examine the type mismatches from the publishers' perspective. We find that there are numerous cases where publishers consistently mistype the type of account they hold within an advertising network. For instance, *mangaread.org* has declared the wrong relationship type for 681 distinct identifiers included in its `ads.txt` files. To give a better understanding, *beachfront.com* has issued the identifier 13310 for an INTERMEDIARY account. However, *mangaread.org* declares in its `ads.txt` file that this identifier is DIRECT. We find 14 websites that repeatedly mis-classify the type of over 600 identifiers in their `ads.txt` files.

> **Finding:** The ads.txt and sellers.json are not properly used with over 26K publisher IDs being declared with the wrong type. This phenomenon is observed for IDs of all ad networks.

## E.2 Misuse

In addition to the discrepancies described in Section E.1, we also discover that there is wrong application of the `sellers.json` standard. First, we discover that there are various domains that copy and serve Google's `sellers.json` file without being affiliated in any way. We find 28 such domains that come from different countries and represent various categories ranging from model agencies, to online shops and business websites. All of these domains serve a copy of Google's `sellers.json` file and inside this file they even provide Google's contact information. We exclude the `sellers.json` files of these domains from any further analysis of Section 5 since they do not represent actual business relationships between websites and ad networks.

In addition to this, we discover multiple cases where entries listed in `sellers.json` files concern domains that the clients most likely do not own, manage or are in any way related. For example, we find that Reklamstore's `sellers.json` file contains over 25 entries

```
{
    "domain": "facebook.com",
    "is_confidential": 0,
    "name": "malik",
    "seller_id": "614c165d57a...",
    "seller_type": "PUBLISHER"
}

{
    "domain": "facebook.com",
    "is_confidential": 0,
    "name": "aliana",
    "seller_id": "3648d3493...",
    "seller_type": "PUBLISHER"
}

{
    "domain": "facebook.com",
    "is_confidential": 0,
    "name": "la casa del árbol",
    "seller_id": "3df3a6197...",
    "seller_type": "PUBLISHER"
}

{
    "domain": "m.facebook.com",
    "is_confidential": 0,
    "name": "Barcelona",
    "seller_id": "4ec7af59f4...",
    "seller_type": "PUBLISHER"
}
```

**Figure 12: Snippets of the `sellers.json` file served by the ad network *adyoulike.com*. There are multiple entries for the *facebook.com* domain, all of which seem to be deceitful.**

for the domain *youtube.com*. It is obvious that a lot of these entries are not valid because they have a PUBLISHER relationship and the registered owner is a YouTube channel or some random names (e.g., "fkt"). In general, we discover that in multiple ad networks, people are able to register popular domains (e.g., *google.com*, *twitter.com*, *facebook.com*) as their own, using their own names. In Figure 12, we illustrate some examples, where multiple accounts have registered Facebook in AdYouLike. This suggests that ad networks don't properly review the information their clients submit, or that this process is highly automated.

Finally, we find that there are almost 10,000 domains which are listed as an INTERMEDIARY in different `sellers.json` files, but these domains do not seem to be ad networks and in fact do not serve a `sellers.json` file themselves. According to the specification [42], when the seller type property is set to INTERMEDIARY, the listed domain should point to the root domain name of the seller's `sellers.json` file. This is not the case for thousands of entries. Even if this listing is done by accident or if some entities simply do not fully adhere to the `sellers.json` standard, the fact is that this behavior deteriorates the ecosystem's transparency and makes the end-to-end verification of involved entities practically impossible. In fact, even Google does not follow this rule and serves its own `sellers.json` file through a different domain (i.e., *http://realtimebidding.google.com/sellers.json*).

> **Finding:** Ad networks pay little attention to their `sellers.json` because users can claim any website they want.

## E.3 Transparency

The `sellers.json` standard is supposed to provide greater transparency to the online advertising ecosystem and a better understanding of how revenue flows across different entities. This is especially useful for advertisers since they can keep track of where their money is going and who they effectively fund. Apart from the ethical aspect, advertisers are eagerly interested in understanding who they fund because they can increase their audience engagement and get a better ROI. However, it is often the case that ad exchanges hide the required information through the confidentiality flag that the `sellers.json` specification describes [42]. In such cases, the ad networks only publish the seller ID and seller type, which are mandatory. Hiding this information makes it impossible for advertisers to quickly understand that their ad spending is

funding specific websites or know which entities are involved in these ad transactions [59].

Towards that extent, we examine all `sellers.json` files in our dataset. Unfortunately, we find that a lot of ad networks do not work towards a more transparent ad ecosystem and that they actively try to hide or obfuscate their operations. Such ad networks have thousands of clients and hide all of their identities in the `sellers.json` file they disclose. For example, `MyTarget`, a Russian ad network, has issued 4,877 distinct identifiers for its clients but has marked all of them as confidential without showing any domain name or owner name. Similarly, `Concept.dk`, `Unibots` and `I-mobile Co.` are all advertising networks with thousands of clients that have completely confidential `sellers.json` files and do not disclose the identity of any of their clients.

To make matters worse, we observe that this behavior is even common among top ad networks that dominate the market. `Adlib`, `adreact` and `adstir` are popular ad networks with a substantial amount of clients and in all cases, more then 94% of their `sellers.json` files are confidential. Similar behavior is observed in Google's `sellers.json` file, which we find is the biggest and more widely used ad network. We observe that, as of March 2023, Google has issued 1,277,156 identifiers, 75.53% of which are confidential. According to their official documentation [33], a lot of their entries are confidential (including the domain) in order to protect the privacy of individual accounts that have registered to the service with their personal name. Nonetheless, this intentional behavior makes the ad ecosystem extremely unclear, thus beating the whole purpose of `sellers.json` files. On the other side of the spectrum, networks such as `GumGum`, `SmileWanted` and `Sublime.xyz` serve completely transparent `sellers.json` files, listing the domains and name of all of their clients.

Altogether, the `sellers.json` standard is regularly misused. According to the specification [42], `sellers.json` files are supposed to increase the transparency of the ad ecosystem and enable the identification of the entities that participate in it. However, it looks like it is not properly enforced and implemented and that both publishers and ad systems are accountable. If the standard is constantly and to a large extent misused, then there is no trust or transparency, and sooner or later bad actors will devise new techniques to abuse the ecosystem and elicit advertiser money.

> **Finding:** The `sellers.json` has not achieved its original goal since it is regularly misused and brings no substantial transparency to ad supply chains.

## F  Hidden Intermediaries Cost

In this section, we provide an estimation of the potential cost that hidden intermediaries can have on the advertising ecosystem. For each of the clients of hidden intermediaries discovered in Section 5, we extract network and demographics data from SimilarWeb [48]. We are able to extract accurate data for 146 client websites of hidden intermediaries. We find that for 64% of these websites, the majority of the visitors come from the United States, an audience with great geographic value for advertising [72].

We attempt to translate the network traffic of these websites into ad revenue, using Google's ad revenue calculator [35]. We map information about the category of website and the country of origin

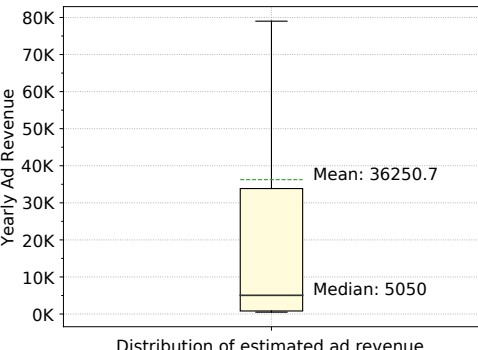

**Figure 13: Distribution of yearly ad revenue of hidden intermediaries clients based on Google's AdSense.**

of its audience to the respective taxonomy system that the revenue calculator tool uses. Additionally, we round network traffic to the closest accepted value if the reported monthly visits are less than the minimum or greater than the maximum accepted value. We plot in Figure 13 the potential yearly earnings from ad revenue for these websites. We discover that on average, a client website can generate 36K USD from ads and that, in total, clients of hidden intermediaries can cost advertisers 5.3M USD.

Please note that the mentioned revenues are simple estimations. Google's tool estimates revenue based on the content category and the location of traffic. The actual revenue of a website can vary greatly based on various features, including the actual ad network that delivers an ad, user demographics (e.g., age and gender) [60], user interests and device type [64], and advertiser demand.

## G  AdSparency: Investigative Tools & Functionality

We develop and publish AdSparency: a Web monitoring service that unveils the business relationships among websites and ad networks, as well as potential revenue flows. AdSparency utilizes information extracted from `ads.txt` and `sellers.json` files, and enables (i) stakeholders (e.g., advertisers, DSPs, Web publishers) to better understand where their money is funneled and what content they support, and (ii) policymakers (e.g., IAB, WFA) to better understand ad revenue flows and business relationships. Specifically, this service periodically crawls millions of domains, aggregates files served by different domains and analyzes the corresponding entries. Then, it (i) provides important statistical information about Identifier Pooling and Hidden Intermediaries, and (ii) provides a collection of investigative tools. Specifically, AdSparency provides tools to (i) lookup identifier pooling, (ii) detect hidden intermediaries, (iii) study website partnerships, and (iv) reveal business relationships among publishers and ad networks. By crawling large sets of websites, this service can zoom out and reveal to marketers, publishers and ad agencies the bigger picture: how publisher IDs are used and what relationships are formed between the various Web entities in a global scale. It should be noted that AdSparency provides a systematic insight into the ad ecosystem without inferring any findings with a makeshift methodology. All of the provided evidence is

reported by the publishers and the ad networks themselves through `ads.txt` and `sellers.json` files.

**Identifier pooling lookup tool.** This tool helps stakeholders understand which publishers share the same "wallet" (i.e., use the same account to aggregate ad revenue). Users provide a specific publisher ID and see which websites explicitly declare this ID as `DIRECT` (i.e., direct control of the account) in their `ads.txt` file. Additionally, users can explore if the same ID is declared as `RESELLER` by other domains, suggesting a discrepancy in the way it is used. This information is acquired from over 81M `ads.txt` entries.

**Tool to detect hidden intermediaries.** This tool aims to disclose the business relationships that ad networks form. Specifically, users are able to query for a domain and discover if this domain has registered as both a `PUBLISHER` and an `INTERMEDIARY` in multiple ad networks. We derive such information from the `sellers.json` files served voluntarily by ad networks, thus increasing our confidence about its correctness. We only report business relationships with unambiguous information. That is, we only process `sellers.json` entries that explicitly state the domain who registered for a specific publisher identifier. Using such information, stakeholders can deduce if specific ad networks display a suspicious behavior by sometimes registering as a content owner (i.e., publisher) and other times as the facilitator of ad impressions (i.e., intermediary). Note that this information cannot constitute concrete evidence of an entity abusing the ad ecosystem, but rather an indication of misuse.

**Tool to examine behavior of websites in terms of partnerships.** This tool aims at enhancing the transparency regarding the relationships or even ownership [62] of websites thus help uncover potential dark pools as they were formed in the past [22]. Specifically, a user can query for a domain and discover with what other websites this domain shares `DIRECT` identifiers.

**Tool to reveal business relationships among publishers and ad networks.** This tool, analyzes `ads.txt` and `sellers.json` entries and presents the relationships a publisher claims to have with various ad networks, as well as the ad networks that claim the provided domain is a registered publisher within their network. Previous work has demonstrated that such information can uncover the facilitation of objectionable content on the Web [63].

**Tool to fetch `ads.txt`/`sellers.json`.** Finally, our service provides modules to fetch and present the `ads.txt` and/or the `sellers.json` files of a specific domain.

