# OpenReview forum: "Welcome to the Dark Side: Analyzing the Revenue Flows of Fraud in the Online Ad Ecosystem"
_ACM.org/TheWebConf/2025/Conference — WWW 2025 Oral_

### Official Review · Reviewer_rzy3 · 2024-11-25

**Novelty:** 6
**Technical Quality:** 6

**Review:**

### Quality:
The study sets itself apart by analyzing a broad dataset encompassing over 7 million websites, ensuring its findings represent the broader digital advertising ecosystem. Employing both quantitative and qualitative methods, such as graph-based analysis and WHOIS records, the authors examine how online advertising standards are misused. This dual-method approach provides both depth and breadth in uncovering systemic fraud mechanisms. Additionally, the authors enhance the research community's ability to validate and extend their work by making their datasets and tools publicly available, reinforcing transparency and promoting further investigation.

### Clarity:
One of the paper's notable strengths is its clarity. The authors provide detailed explanations of technical terms and mechanisms, such as ads.txt and sellers.json, ensuring that readers unfamiliar with these standards can follow the discussion. Additionally, the inclusion and use of visualization techniques such as graphs and tables help better understand the distribution of identifier pooling and flow of revenue.

### Originality:
The study makes a contribution to the field based on the large-scale systematic investigation and study conducted into the ad transparency standards. The discussion of well-known methodologies in combination with the introduction of AdSparency (a monitoring tool for revenue flow visualization and unearthing business relationships - illegal or not) adds to the originality of the paper. Furthermore, the authors showcase extensive research work in the area of fraudulent ad activities by addressing identifier pooling and hidden intermediaries.

### Significance:
The major significance of this study is the ability to showcase the systemic fraudulent issues within the online advertising ecosystem, especially in ways that enable fake news and piracy. By bringing this to the forefront, it creates implications for not just advertisers but publishers, regulators, and policymakers. In addition, the decision by the authors to make available publicly the dataset and tools used for this paper created a potential industry-wide drive to inspire changes and further research efforts.

### Pros and Cons
#### Pros:
1. Vast and comprehensive dataset (>7 million websites).
2. Identifier pooling and hidden intermediaries.
3. Promotion of novel tools, transparency, and further research (AdSparency).

#### Cons
1. Reliance on Voluntary compliance by stakeholders.
2. Structural issues that drive fraudulent practices within the advertising ecosystem are not addressed.
3. The differences between legitimate and abusive identifier pooling are not clarified well enough to prevent possible confusion on the part of the reader.

**Questions:**

1. The paper highlights significant issues with identifier pooling but notes that not all identifier sharing is inherently abusive. Could the authors elaborate on the specific criteria used to distinguish legitimate pooling (e.g., between entities owned by the same organization) from abusive or fraudulent pooling? How were edge cases handled in the analysis?

2. While the paper proposes two countermeasures, what specific steps would the authors recommend for policymakers or regulatory bodies to enforce ad transparency and reduce ad fraud? How feasible is it to mandate compliance with enhanced standards (e.g., stricter ads.txt definitions) across a fragmented global advertising ecosystem?

**Reviewer Confidence:**

4: The reviewer is certain that the evaluation is correct and very familiar with the relevant literature

**Scope:**

4: The work is relevant to the Web and to the track, and is of broad interest to the community

---

### Official Review · Reviewer_DUS9 · 2024-11-28

**Novelty:** 6
**Technical Quality:** 7

**Review:**

This paper investigates the pervasive issue of fraud within the online advertising ecosystem. It analyzes the abuse of ad transparency standards by "identifier pooling," i.e. when fraudulent websites use legitimate publishers' identifiers to siphon ad revenue.  The paper has two key contributions as I see it. First, a large-scale study of over 7 million websites. Their results indicate that dodgy websites frequently engage in this id pooling (!). Second, a practical solution that authors name "AdSparency."

The presentation of the paper is generally strong. The writing style is clear and concise, with a good flow of ideas. I personally wonder whether it would be clearer to introduce ad pooling from a top-down perspective rather than a bottom-up perspective. But that's too minor to worry about.

I have one small remark that I disagree with the authors when they state that private WHOIS records do not affect the correctness of our findings, making them representative of the Web. That is not true! Your findings are lower bound and not representative!

Overall, I believe this is a very strong paper. It identifies a relevant problem (or atleast, identifies that a already known problem is even bigger than previously thought), conducts a thorough quantitative analysis of the issue, and offers a solution. I think this is a valuable contribution to the WWW community.

**Questions:**

- Can you justify the word representative mentioned above?
- Could you clarify where do you come up with the 15x prevalence difference? What figures are you comparing?

**Reviewer Confidence:**

3: The reviewer is confident but not certain that the evaluation is correct

**Scope:**

4: The work is relevant to the Web and to the track, and is of broad interest to the community

---

### Official Review · Reviewer_pPph · 2024-12-01

**Novelty:** 6
**Technical Quality:** 5

**Review:**

This paper discusses the issues with online ad ecosystem, especially the issues caused with / hides behind advertiser ID sharing, pooling and hidden intermediaries. It contains many interesting and potentially important findings; the author(s) performed large amount of analysis; the techniques are sound in general. The reviewer is not familiar enough with ad ecosystem to reliably evaluate the novelty of the work, but is satisfied by the relevant discussion in the paper. However, there are some issues with the writing (the story aspect, not the grammar aspect), leading to difficulty to evaluate part of the paper's claims and story as a whole.

**Pro**: Extensive analysis is performed in the paper, and the discussions covered many important aspects about the implications of those findings from statistics, and their real-life importance.

**Con**: Writing style is not appropriate, at least to anyone outside of the narrow community. Important background information or relevant explanation is missing, for both the paper story itself, and the figures, leading to the difficulty to evaluate part of the claims. Abbreviations and methods are used without explanation.

I am writing down my most important comments (including questions) here, with a good faith, in the hope that the author(s) can explain and/or justify in the paper to improve the quality.

C1: Jargon and confusion
------------------------

As a CS researcher from a different field, I find it extremely difficult to understand the different jargon about ad ecosystem, and the relation between different actors. There is no explanation of them in the paper, nor pointers to explanations, making it hard to understand what to focus on. It appears the author also used some terms as synonyms (esp. Sec 5), increasing the difficulty to understand. Also, there are some abbreviations without explanation.

In general, this is the biggest complaint I have. I do appreciate that the paper has packed significant amount of experiments and findings, but that will only be useful if the readers can understand.

I have noted down the terms I still find confusing at the second half of the paper:

*   ad inventory
*   inventory
*   publisher
*   client
*   user
*   ad network
*   ad system
*   seller
*   Line 596 - 603 mentioned terms like “companies”, “publishers”, and “intermediaries”. What do they mean? In particular, who are these parties, and how do they relate to the “websites”, “ad networks” and other terms used in Section 4?
*   CPM

The list is nowhere near complete complete, and I actually struggled a lot in the first half of the paper. To be honest, that almost caused me as a reader to give up.

C2: Insufficient background information
---------------------------------------

The Background section only provided very limited information. Combined with the issue above, it is hard to understand the description. In addition, is is unclear why the described two documents are so important to be the only information in Background section.

C3: Missing and/or confusing technical details
----------------------------------------------

*   For after Line 312, when constructing the pools, what do you do with multi-to-multi mappings? For example, publisher ID `A` already forms a pool of 5 websites, and publisher ID `B` forms a poll of 6 websites. Then, at a new website, it lists both `A` and `B` in its `ads.txt` file. How do you form the pools? Combine all of them together as a big poll of `5+6+1=12` sites? Or include the new site into both pools, but not form a new pool?
*   Line 337 says ad networks not properly using `ads.txt`, but it is not clear why is this not proper. In fact, in a few lines above, Line 316 - 318, the author said it is not an issue for the same entity to share identifiers across their websites.
*   Related to the above, does the publisher ID correspond to first-level domains (e.g. `my-site.com`) only, or any level of domains (e.g. `a.b.my-site.com`)? Is it expected that each website (for a certain level of domain) has one publisher ID?

C4: Lack of clarify of information
----------------------------------

*   Why does Figure 6 plotted as CDF (Cummulative Distribution Function? Unexplain abbreviation), if the author wants to also draw attention to median and mean?
*   Why does Figure 6 only compare fake news pools versus all pools? How about piracy pools?
*   Line 388 mentioned Kolmogorov–Smirnov tests, but without citation or explanation on what it will do, why this test is selected, and how to interpret the results. Although I do partially understand, it would be better if this can be clarified.
*   In the paper, the p-value is often (if not all) represented in the form of `5.0815e-320`. It does not make too much sense to give this details as the p-value is so tiny; instead, saying p<.001 should be sufficient. If really needing to report the value, it would be better to use the normal scientific notation to avoid any confusion, as the e-notation is not universal, and papers are not restricted to simple string texts.
*   What does “the ecosystem one studies.” mean on line 459?
*   What does “popularity” mean for Figure 6 and afterwards (e.g. Line 479-487)? Is it the number of visits per day/month/year? Or is it the order of ranking in Tranco list
*   What does “popularity cut-off point” mean? at Line 493 - 494? It is hard to tell what Figure 7 tries to convey, especially what does the Y axis represent, and how that relates to the discussion between Line 494 - 501? There is indeed a turning point, between ~0.5M to ~1.5M. But what is the criteria for a plateau, making the author claiming it is after 1M? For example, why is it not after 3M? Also, how do the reader know that the increase of a point is less than 1% from the figure? By reading the y-value? But the y-values don't add up to `1`, thus they cannot be the percentage of contribution, isn't it?

C5
--

After reading the paper, a question is still not answered thus affecting my evaluation of the importance of the findings. That is: is the finding (esp. Sec 4) particular for fake news and piracy websites? I know Figure 5 provides some differences between “all” pools and these two special types of pools, but that is just the size, and does not seem to apply to the further discussions (e.g. Figure 6 - 8). Is that simply not possible (or too difficult), or due to other reasons?

C6
--

I'm having difficulty in understanding and evaluating the discussion between Line 541 - 551. It appears the author attempted to discuss how much revenue such pools may make, to discuss the potential harm the such pooling may bring. However, it is not clear that:

1.  Why SimilarWeb was used for getting the visit data?
2.  Line 548 said 16 websites, but why these 16 websites? In particular, Line 519 said there were 93 + 31 = 124 **pools** (of fake news and piracy) for rank<20K pools, but why only 16 **websites** here?
3.  How is the revenue extracted? What is the source or method?

C7
--

Line 564 - 566 mentioned “extreme misinformation websites”, which presumably refers to fake news websites. Why is there a term switch, and why is this switch correct?

C8
--

Line 564 - 567 talked about the discovery from this study, which I do not disagree. However, is it not possible for such websites to generate revenue if they do not use such pooling methods? Can they not directly obtain an advertiser ID from ad networks?

C9
--

I find Line 635 - 652  confusing as well, due to lack of background information and explanation.

What is the relation between an “ad system” and an “ad network”? Are they equivalent? If so, I am curious if there is any rationale to focus on only 10 ad networks out of 2682 in Section 4? I understand they are the top 10, but is there a reason to cut at 10, or is it simply a choice by the author?

Why is Line 645 talking about `ads.txt`? How does that relate to the discussion about `sellers.json`?

Line 648 says “users”, but who is a user? Isn't the `sellers.json` published by ad network or ad system (owner)?

C10
---

I attempted to understand the information given between Line 655 - 659. The following needs the author's clarification:

1.  Can an entity X publish both `ads.txt` and `sellers.json`?
2.  Why is X called an “ad network”? Does the term "ad network" refer to a) anyone providing ads, regardless of the source of the ads, or b) anyone allowing advertisers to sell ads, and then provide ads to content publishers? It appears both definitions don't fit into this sentence.
3.  Why do these four conditions all need to meet for X to be a hidden intermediary? In particular, why does X have to serve a valid `sellers.json` file to be a hidden intermediary? They want to be “hidden intermediary”, so why would they disclose that they are an intermediary by having `sellers.json`?

C11
---

The paper mentioned several times that advertisers won't want their ads on questionable / illegal content websites, e.g. Line 757. However, is there any evidence supporting this claim? I did not seem to find it in the paper.

C12
---

I understand it is not ideal and probably rule-breaking for sharing your advertiser ID with other sites; piracy and fake news are also generally known to causes problems. But why is sharing advertiser ID with piracy, fake news or other similar forms of websites particularly a problem? Especially, what **harm** is caused by that?

**Questions:**

I am aware some of the questions above in fact correlate to each other, so I'm not listing them all here. Instead, here are the most important ones:

1. Who are the relevant actors and what are their relations in the ad ecosystem, where they are mentioned / relevant to the paper? (C1)
2. Why only the two documents are mentioned in background section? (C2)
3. Can the author please answer the relevant questions in C3?
4. Are there any comments on the last question in C4, about Figure 7?
5. Any comments to C5?
6. Any comments to C8?
7. Any comments to C12?
8. Any comments to other items, especially if there is any misunderstanding?

**Reviewer Confidence:**

3: The reviewer is confident but not certain that the evaluation is correct

**Scope:**

4: The work is relevant to the Web and to the track, and is of broad interest to the community

---

### Official Review · Reviewer_fGbX · 2024-12-01

**Novelty:** 5
**Technical Quality:** 5

**Review:**

Paper summary: The authors explore a critical problem in digital advertising -- how malicious actors bypass ad network restrictions due to the opaque supply. This study of sellers.json and ads.txt shows that publishers of questionable or illegal content inflate revenue by pooling ad identifiers with reputable sites. Additionally, the authors reveal misuse of the sellers.json standard, where intermediaries pose as publishers to divert ad revenue to objectionable websites. Their findings highlight that current standards like ads.txt and sellers.json are insufficient to prevent ad fraud.

Strength:
1.	The authors conduct extensive large-scale analysis to examine the current ad transparency standards. The findings are interesting. The released code and data is a valuable resource to the community.
2.	This presentation (especially the figures) is great. The tables and figures convey informative details. For example, Figure 9 shows the flow vividly.

Weakness:
1.	Fraud-related research has been studied for years. This topic is not that interesting or exciting. The motivation of this work may not be strong enough.
2.	Maybe I miss it in the paper. I have clarification questions here. How do we define and identify the ground-truth bad actors? How can transparency be quantitatively evaluated by metrics, given that transparency is a significant concept across the paper?
3.	I am wondering about the time dimension of the analysis. How the findings and the standards change over time. When new data comes, will the finding still hold?
4.	Since there are many sections and findings, users can get lost. What is the takeaway message?

**Questions:**

Please see the mentioned weakness.

**Reviewer Confidence:**

3: The reviewer is confident but not certain that the evaluation is correct

**Scope:**

3: The work is somewhat relevant to the Web and to the track, and is of narrow interest to a sub-community

---

### Official Review · Reviewer_Yf8u · 2024-12-03

**Novelty:** 5
**Technical Quality:** 5

**Review:**

The paper provides a thorough and rigorous exploration of fraudulent practices within the online advertising ecosystem, particularly focusing on the misuse of transparency standards like ads.txt and sellers.json. The large-scale dataset of over 7 million websites, combined with systematic analysis and graph-based methodologies, offers compelling evidence of how bad actors exploit gaps in these standards. Notably, the study uncovers underreported issues such as identifier pooling and hidden intermediaries, presenting a complex understanding of the scale and mechanisms of ad fraud.

The work is both original and significant, as it extends prior research by an order of magnitude in scope and introduces AdSparency, a practical tool designed to enhance transparency in the ad ecosystem. The findings have immediate relevance for advertisers, policymakers, and researchers aiming to combat fraud and improve accountability. However, the revenue analysis relies on assumptions that could affect its accuracy, and the geographical scope of the dataset is not sufficiently discussed, potentially limiting the generalizability of the conclusions.

Overall, the paper is a valuable contribution to the field of internet technologies and digital advertising. Its insights into systemic vulnerabilities, coupled with actionable recommendations and an open science approach, make it a critical resource for addressing the challenges of ad fraud. However, further evaluation of the AdSparency tool in real-world settings and broader contextual analysis would strengthen the impact of this research.

**Questions:**

Data Validity: How did you handle false positives and negatives in identifying fraudulent identifiers and hidden intermediaries? Could inaccuracies affect the robustness of your conclusions?

Legitimacy of Identifier Pooling: What criteria did you use to distinguish between legitimate and fraudulent identifier pooling, especially for organizations managing multiple domains?

Revenue Estimation Assumptions: What assumptions underlie your revenue estimates for fake news and piracy websites, and how sensitive are the results to these assumptions?

Sellers.json Adoption Bias: How does uneven adoption of sellers.json files across networks impact the completeness and accuracy of your findings?

Temporal Adaptation: Did you observe shifts in fraud patterns over time, and if so, how might these trends affect the generalizability of your results?

**Reviewer Confidence:**

4: The reviewer is certain that the evaluation is correct and very familiar with the relevant literature

**Scope:**

4: The work is relevant to the Web and to the track, and is of broad interest to the community